# Sustainable Approach to the Development of the Tourism Sector in the Conditions of Global Challenges

Petar Bojović [1], Aleksandra Vujko [2,*], Miroslav Knežević [2] and Radmila Bojović [1]

1    Independent Researcher, 11000 Belgrade, Serbia; p_bojovic@yahoo.com (P.B.); radmilab@yahoo.com (R.B.)
2    Faculty of Tourism and Hospitality Management, Singidunum University, 11000 Belgrade, Serbia; mknezevic@singidunum.ac.rs
*    Correspondence: avujko@singidunum.ac.rs

**Abstract:** The main objective of this study is to present the feasibility of rerouting the EuroVelo 6 through the Fruška Gora National Park (Serbia) as a sustainable eco-tourism product. Our research was conducted with a survey and field interviews. The first part of this research involved consulting 717 cyclists from six EU countries, and the opinions of local entrepreneurs and employees of the national park were also investigated. An architect made a "blueprint" info board as a proposal for trails within the national park. The purpose of conducting this research was to show the importance of retracing an international route. Cyclists expressed their satisfaction with the proposed F1 track to the greatest extent. The results of the survey of employees of the NP suggested the sustainability of cycling tourists because they showed great responsibility in their behavior, and the results of the domestic entrepreneurs particularly pointed to some economic aspects of the development of cycling, as well as its impact on the empowerment of women in rural destinations. It was concluded that rerouting the "Danube Cycle Route" through the Fruška Gora National Park is feasible and sustainable. The results obtained in this study provide a justification for this retracing because it turns out that cyclists are desirable, responsible, and sustainable tourists who have a significant impact on the destinations and people they are directed to. This research will be applied to every attractive part along EuroVelo 6, thus expanding the network of cycle paths.

**Keywords:** tourism business; sustainability; forms of tourism; sustainable development; greening of tourism; ecological aspects; synergy effect

## 1. Introduction

Cycling tourism is much more than moving between points. This form of tourism represents a "philosophy" of movement, where movement is much more than a transport concept and travel is much more than a tourism concept. This statement is supported by the works of numerous scientists and theorists who have researched various aspects of cycling in the last decade. One of the most common approaches to cycling is health, but in this paper, we will not dwell on this aspect much. The extent to which cycling promotes health has been studied and is well established [1–3]. For example, in his study, Unwin [4] found that health improvement was observed in British people as a result of cycling. As they used cycles more and more, it was observed that male civil servants who were regular cyclists (those who cycled for at least an hour every week) had a coronary attack rate that was less than half of that of non-cyclists. Mulley et al. [5] and Deenihan and Caulfield [6] present another method by which they proved the health benefits of cycling. They based their research on the benefits of cycling in the prevention of mortality. According to Mulley et al. [5] and Deenihan and Caulfield [6], those that cycle can reduce their risk of mortality by 16%. In addition, there are numerous studies that support the fact that cycling is one of the best ways to fight and prevent obesity [7,8], hypertension [1], and stress [8,9]. According to Pucher and Buehler [10], European countries such as Germany

and Denmark have a lower percentage of obese people than, say, North America, where the number of those who are dependent on cars is many times greater. Therefore, many scientists agree that physical inactivity is the reason for many diseases with fatal outcomes and that cycling is "the best medicine" [2,11,12]. There are data from Denmark, one of the leading countries when it comes to cycling in Europe [10], that indicate that cycling is estimated to prevent about 3000 deaths, >3000 cases of type 2 diabetes, almost 6000 cases of cardiovascular diseases, and more than 2000 cases of cancer, per year [12,13]. This and many other examples [8,14,15] prove that cycling contributes to physical well-being [16]. In addition to this, there is also evidence of emotional wellness benefits, as it can be seen that cycling has therapeutic properties because riding through nature, especially national parks, directly affects mental health and relaxation [17–21].

In addition to health, cycling has other aspects that are much less studied in theory, which represent the main motives of most countries to contribute to cycling development. These are ecological and economic aspects of development (E2). This paper summarizes both of these aspects of cycling, with the ultimate goal of presenting the project of rerouting the "Danube Cycle Route" (EuroVelo 6) through the Fruška Gora National Park (Serbia) to include the entire national park in the project of networking the area with a system of cycle paths. The purpose of this work is to promote a healthy way of moving through space and, at the same time, show the importance of retracing an international cycling route such as EuroVelo 6. It is a fact that this kind of cycling route, specifically, an international route, affects numerous cross-border partnerships because the Fruška National Park Gora is located in the border area that separates Croatia (EU) and Serbia, which has the status of a pre-accession member state. Investing in the construction of the necessary signage with respect to cycling would also affect rural development. Through our research, we wanted to show that all these aspects have positive effects on the local population because developing cycling tourism in rural destinations would contribute to the opening of new workplaces and also the empowerment of women, which is a particularly sensitive topic.

This work started with the main notion that rerouting the EuroVelo 6 route is desirable, feasible, and sustainable, and as shown in other papers, cyclists prefer to cycle through nature and forests rather than on concrete paths. This is especially the case in the summer months when temperature differences are noticeable [17,22,23].

Cycling is a preferred way of moving through space, and cycling tourists are a preferred category of tourists. It is very important to understand the "legalities" that affect the development of cycling tourism in a destination, as these impact the expansion of official international routes such as EuroVelo 6. Numerous attractions are located along the mentioned route, but cyclists do not know about them and thus pass them by. Visiting tourist attractions in the immediate vicinity of the routes would bring two-way benefits for both cyclists and the countries through which the route runs. First, this means that cycling tourists would have longer stays at the destination and, therefore, heighten their consumption. On the other hand, cycling tourists will get to know the tourist attractions at the destinations and thus enjoy a more complete experience along the route. There are not many studies in the literature that deal with the retracing of bicycle routes in the world, but there are studies that talk about the importance of cycling tourism development in certain destinations. Therefore, the present work will make a unique contribution to the scientific literature. The authors hope that this work will encourage the decision-makers responsible for the Fruška Gora National Park, as well as those in the official Ministry of the Republic of Serbia, to invest in projects to expand the "Danube Bicycle Route" and promote the development of cycling tourism as a sustainable, responsible, and desirable form of tourism.

### 1.1. Ecology and Cycling

From an ecological point of view, cycling is closely linked to the term "ecotourism". It is about "responsible" travel and the use of sustainable means of transportation along the way. Therefore, it is about destinations such as protected localities and national parks,

where the main attractions are natural attractions. From this, we can conclude that such destinations require controlled movement without pollution that does not harm biodiversity. In science, the term "caring capacity" [24] is used to define the limited number of users in a given space, but natural attractions attract tourists, so there are always more of them in such destinations. Even with a limited number of visitors, consideration must be given to where and how tourists move around. The best way to accomplish this is to mark the routes. A system of marked paths through a national park makes it easy to move around and visit all the secured and accessible sites. Such routes are beneficial for both the national park and the tourists. Controlled movement protects the plants and animals of the national park and makes tourists feel safe. The bicycle, as a means of transportation, is an ideal way to get around in national parks. The advantages include the reduction in noise and air pollution [25,26]. The only energy required when cycling is provided directly by the traveler, and the use of this energy alone represents a valuable cardiovascular workout, as we have already discussed. In addition, bicycles also require much less parking space, so the indirect costs to public infrastructure are also very low. Countries where cycling is developed, such as the Netherlands, Denmark, and Germany, serve as an example of what good practice should look like. The Netherlands, Denmark, and Germany are among the most successful countries in promoting cycling [27]. All three countries have a very high standard of living, and car ownership in the three countries is among the highest in the world [10]. This means that awareness of the importance of cycling and the benefits that cycling has for the individual and the environment has been raised among these people. A successful cycling policy led by these countries should provide useful guidance on how to raise awareness of cycling in other countries. These routes are not only the backbone of tourism development in the areas they pass through but also the promoters of a lifestyle that is undoubtedly on the rise. The fast pace of life and everything that surrounds modern man leads him to be massively concerned about himself, his recovery, and his prevention, as a prerequisite for "health". A return to nature, healthy and organic food, and spending time with family and friends form the basis of "ecotourism". Everything that strengthens us and makes us healthier, that makes us more powerful and courageous, that inspires us to be better as human beings, and that tends to last because it is based on the right values, is a prerequisite for "sustainable". Only responsible behavior will lead humanity on a path where the future is still green.

*1.2. Economy and Cycling*

In addition to improving health and influencing air quality or the carbon footprint, cycling creates "hard economic value and jobs". Blondiau et al. [28] presented a case study initiated by the ECF (European Cycling Federation) and the Cycling Industry Club (CIC) and carried out by Transport & Mobility Leuven (TML), a consultancy firm from Belgium. Their task was to identify jobs in the European cycling sector. The jobs were identified in five sub-sectors: bicycle retail (mainly sales and repair), the cycling industry (manufacturing and wholesale), cycling infrastructure, cycling tourism (accommodation and restaurants), and cycling services. In addition, the study detailed that more than 650,000 jobs in the EU-27 are directly linked to cycling and that a further 400,000 new jobs are expected in the future. The second part of the study conducted by Blondiau et al. [28] also showed the nature of those jobs. Cycling creates more local jobs and more jobs that require low-skilled labor. Researchers agree that cyclists are more likely to visit local stores, souvenir stores, restaurants, and cafés than tourists using other modes of transportation [29,30]. According to Blondiau et al. [28], the number of cycling tourists is predicted to increase over the next 10 years. For the purposes of this paper, forecasts are presented for the countries from which the largest proportion of cycle tourists originate. These include Germany, which could have a growth scenario from the current 13% to 25.6%; Hungary, from 18.9% to 32.4%; the Netherlands, from 31.0% to 37.3%; Austria, from 8.1% to 18.2%; Romania, from 5.0% to 12.5%; and Slovenia, from 7.1% to 16.5%. This would provide an economic boost to city centers, towns, and villages across Europe.

Applying this type of calculation to Serbia would mean that there are more cyclists and, therefore, higher spending on all items. Some other studies conducted by researchers support this thesis [31]. Cycling tourists spend less per store visit, but they frequently stop at different shops, stores, viewpoints, and restaurants [32,33]. Furthermore, the economic indicators for the development of cycling in the EU are also reflected in the projects. The Interreg program, for example, helps to fund transnational and cross-border cycle path projects. In agreement with the European Cyclists' Federation (ECF), the EU funds Eu-roVelo programs that connect and network long-distance cycle routes throughout Europe [34]. The EU distributes financial resources and opens funds for countries that are not members of the EU in order to include countries that do not have these financial resources in the cycle path system. What speaks in Serbia's favor is the fact that the EU, in cooperation with the European Cyclists' Federation (ECF), examines projects for the expansion of official routes and the rerouting of the existing ones every year [35]. The EU also promotes research into cycling and the exchange of information on best practices among EU countries, just as national governments do within individual countries. The development of cycling therefore has numerous benefits [36], and the proper presentation and thorough exploration of these benefits would include many destinations in the branched system of European cycle routes known as the European Cycle Transversal (EuroVelo). This is especially true for places located directly on marked routes [37–39], which means that one of the main tasks in promoting cycling in Europe would be the rerouting of marked routes. The rerouting of one such route (EuroVelo 6) through Serbia is the subject of this paper.

*1.3. EuroVelo Transversal*

There are 19 EuroVelo transversals that link the cycling routes in Europe. Three EuroVelo routes pass through Serbia: EuroVelo 6 (the Atlantic–Black Sea Route), EuroVelo 11 (the Eastern Europe Route), and EuroVelo 13 (the Iron Curtain Route). EuroVelo 6, or as it is also called, the "Danube Cycle Route", starts in Serbia in the far north of the country, near Bački Breg, and follows the meandering course of the Danube all the way, passing or touching some of the better-known and lesser-known natural and man-made tourist attractions along the way (Apatin, Petrovaradin with the Petrovaradin Fortress, Fruška Gora National Park, Belgrade, Đerdap National Park, and others). The total length of the "Danube Cycle Route" through Serbia is around 667 km and is divided into seven stages. At the junction of the second, third, and fourth stages is the Fruška Gora National Park, which represents the spatial coverage of this study, the subject of which is the rerouting of the EuroVelo 6 route through Serbia. In 2019, 755,000 cyclists traveled the EuroVelo 6 route from the Bavarian city of Passau to Vienna, and 10,000 cyclists travel through Serbia on this route annually.

## 2. Materials and Methods

For the purpose of this work, three field surveys were conducted, and a feasibility study was presented for the F1 cycling route, the first of a total of 8 (F1–F8) divided into 4 (A1–A4) zones, as is the planned number of routes in the Fruška Gora National Park. Fruška Gora National Park is a mountainous area in the northern part of Srem (southwestern Vojvodina, Serbia), i.e., on the southeastern edge of the vast Pannonian Plain. This means that most of the national park is located in Serbia and only a small part, in the far west, in Croatia (EU). In its west–east direction, it has a length of about 80 km [38,39]. The route of the "Danube Cycle Route", which would pass through the Fruška Gora National Park and would be almost 600 km long, would be divided into 4 specific zones, i.e., 8 paths. These trails would include the main cultural and natural attractions of the national park, and these trails would also allow cyclists cycling through Serbia to enjoy the sights that the national park offers. At the beginning of the project, the first zone A1 would be rerouted, i.e., 3 paths within the zone with a total length of 170 km. At the end of the project, the mountain would be completely rerouted. Both the citizens of Serbia and foreign cyclists would benefit from this. The first route F1 of the first zone would be rerouted first. It would include the part from Croatia–Ilok via Šid, Erdevik, and Sviloš and pass through the central,

most beautiful part of the national park, over the highest peak Crveni čot (539 m), and the part towards Irig, which is on the right side of the path, at the Banstol crossroads, where it would connect with the regular EuroVelo 6 route.

An architect, one of the authors of the work, was hired for the purpose of design. The idea was to await the participants of the "Danube Cycle Route" at the junction of the official route and the national park. This point is located not far from the border crossing Ilok (Croatia)–Šid (Serbia). The Vojvodina Cyclists' Association was contacted, and the aim of this work was explained to them. This study lasted two years, during the busiest months of the route, from May to October (2022–2023). In total, 717 respondents from six countries (Germany, Hungary, the Netherlands, Austria, Romania, and Slovenia) took part in this study. The cyclists rode along the retraced "Danube Cycle Route" with the help of professional guides and cyclists and were asked to evaluate the route they were riding in detail. The total length of the F1 route they cycled was approximately 70 km, and the entire length was cycled in one day, with stops along the way. Respondents were asked to rate the quality of the route they had ridden and make suggestions for its improvement. We were also interested in how satisfied they were with the experience and whether they would recommend this section to other participants of the "Danube Cycle Route". Another study dealt with the ranger service of the national park rangers. In the public enterprise Fruška Gora National Park, 16 rangers of the protected area work according to the classification prescribed by the Law on Nature Protection, the Law on Forests, and the Law on Game and Hunting of the Republic of Serbia. The guards of the protected area are obliged to apply the Law on Nature Protection, and since they work in the national park, they are also obliged to apply the Law on National Parks. Their rights and duties are clearly and precisely defined in certain articles of the law, including: "Monitoring the movements and activities of all visitors and users of the protected area". For the purposes of this work, an interview was conducted with 9 guards of the protected area, from whom we wanted to obtain data on whether any irregularities had been observed or any nuisance or pollution had been detected along the road on which the cyclists were moving. The third survey concerned business people, i.e., all business services that rely on the retraced F1 route used by cyclists. We wanted to find out to what extent sales in local stores have increased, whether there is an economic difference in the villages through which the F1 passes, and whether this is visible among local business people. We wanted to test whether the hypothesis put forward by Blondiau et al. (2016), i.e., that cycling creates more local jobs and more jobs requiring low-skilled labor, is true. We were interested in which jobs these are and who performs them. Our research using the interview technique was conducted with employees of 5 local stores, a mountain lodge, a wine house, two local restaurants, a hotel at the Iriški Venac picnic site, and a tourist information center. A total of 13 employees were interviewed. The interviewees were questioned on several occasions, and the answers were recorded in writing. Some of the questions asked of the interviewees were "do they record more traffic in their facilities"; "is it possible to hear a foreign language in their facilities during the summer months"; and "do they need to hire an additional labor force" and if so, "who do they hire"? Our conversation with the owner of one of the local restaurants at the Banstol Ethnohaus in Banstol stood out. The owner is also the president of the women's association "Banstolka", which deals with traditional handicrafts. As her restaurant is located at the junction of the retraced and standard Danube cycle route and is a popular resting place for cyclists, we received valuable first-hand information.

## 3. Results and Discussion

There are 22 important points on the F1 route where large boards with a map should be placed. Important points should be marked on the maps, and, above all, the largest marked point should be the place where this board is located. The maps should also indicate whether a path is suitable for a specific type of cycling and for which location (e.g., draw a silhouette of a bicycle and next to it, draw a table indicating that it is a recreational cycle path). At junctions and on paths, signposts should be located on both sides. Restaurants,

monasteries, stables, and mountain huts should also be marked. Routes F2, F3, F4, and others should be marked with a different color. When studying route F1, the architect marked about 120 places where signs should be placed. If the price for the smaller signs is around EUR 100–120, then larger signs with a map would certainly not cost less than EUR 300. This would therefore mean that at least EUR 30,000 would be needed for the signs. A surcharge of 30% is levied on this price due to possible changes, which should therefore also be taken into account. Road marking and filling should be carried out by authorized persons who can participate in the tender. We also recommend installing handrails for pedestrians in some steeper places where some cannot cycle but push bicycles (elderly people or families with children). It is also necessary to create maps, set up information points, and create places for SOS calls. The path should be at least 2 m wide. If it is to be backfilled, the cost of excavation, transporting, and depositing the earth is EUR 7–8 per cubic meter. If slag or mortar is used for the path, then the price with transport and rolling in is EUR 15–18 per cubic meter. The depth of the excavation is at least 20 cm, while the thickness of the soil or slag layer is at least 10 cm. This means that 0.6 cubic meters of material are required for one meter of a 2 m wide road. The construction or completion of paths is not necessary everywhere along the 70 km route, so the price should also be measured in relation to the required length.

The map shows (Figure 1) the F1 route from Vizić to Velika Remeta, which is about 50 km long as the crow flies. These suggestions for the route refer to this length and not to the actual route, which is much longer in reality due to the ascents and descents and the bends. Due to the different states of health and ages of cyclists and other users of this route, it is necessary to provide more regular resting places (at intervals of about 5 km) where boards are placed with information about the difficulty of the next and previous stages and their lengths, the position of the board itself on the total length of the route, and general information about the resort at that location. The boards should be defined as accents in the space in which they are located because if they blend into the space, they would lose their function and visibility, and they should also be illuminated. The resort would be responsible for the maintenance of the boards once they are installed, as they would also serve as advertising. The bottom right corner of the board would be available for interaction and could be changed as desired by the resort. The rest of the appearance would be unchanging and remain the same on all boards.

These frequent rest areas and stops are provided primarily for safety reasons, as this route is accessible to all people, whether they are young, old, or in good physical condition or not. Since there is not a single resort on the first thirty kilometers (with the exception of the forest near Grgurevac), there is a possibility of building one (although this is an area of the national park, and the general development plan provides for building land every 5 km or so along this route), which would contribute to the realization of the planning of this stage. There are points along the route that are of particular importance to visitors, such as monasteries, health routes, and viewpoints, which should be marked with smaller signs, next to which there should be a place to park bicycles.

Six new resorts must be built on the first 30 km. Initially, these could be temporary structures, e.g., containers with a façade designed to fit into the space (e.g., clad in wood). Each resort should have the necessary infrastructure (water, electricity, toilets) as well as first aid kits and telephone lines. The rest areas should be located in different places: Vizić, which would be the first resting place in Serbia for cyclists coming from Croatia; Djipša, where temporary buildings would be erected on agricultural land until a more permanent solution is reached with the municipality; Divoš, which has plenty of building land along the route itself; Ležimir, whose building land is ten meters away from the route itself and can be used as in Divoš; and at kilometer 20 Popov Čot or at kilometer 22 Summer stage, which are two places of interest for visiting cyclists as well as other tourists. At the 28th kilometer from the start of the route, there is a derelict building that could be converted into a hotel (motel), as its construction is still in good condition. Some of the resorts should be ethno-restaurants with minimal accommodation and the possibility of providing first aid.

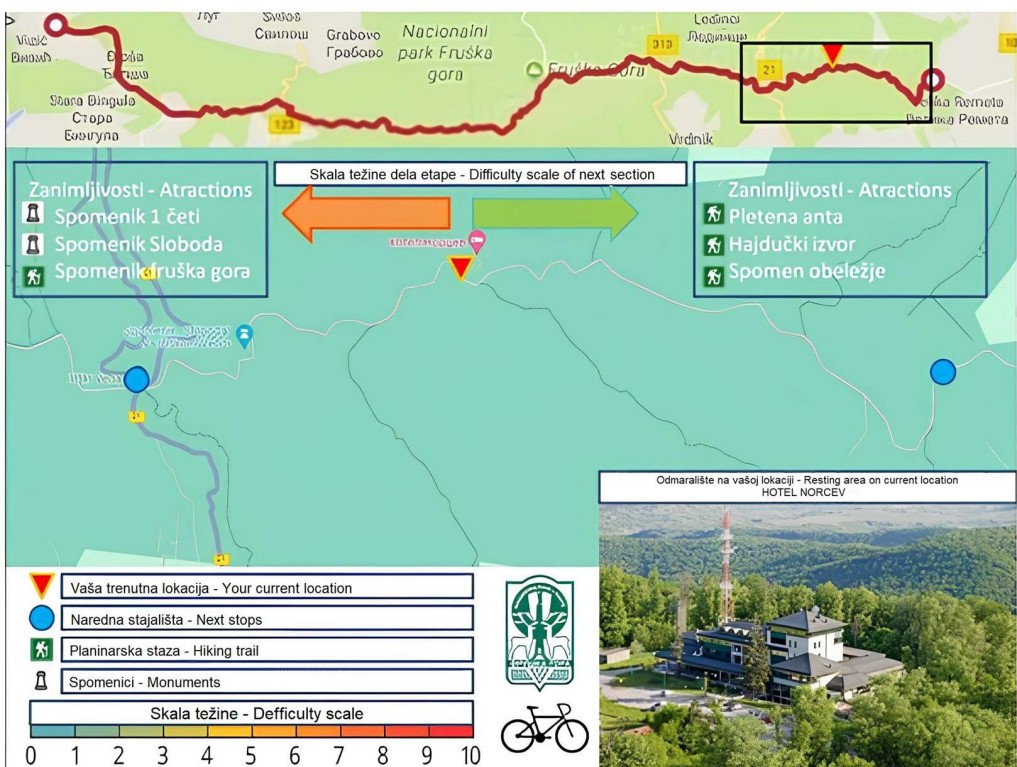

**Figure 1.** Map of cycling route F1 on the Fruška Gora National Park (source: authors).

At about kilometer 35, above Vrdnik and along the route itself, there is a forest recreational tourist complex and a tourist recreation complex on a building plot that could be used as another resort with minimal intervention at this stage. Iriški Venac, which is the center of this stage, is located at about kilometer 40. There are a large number of resorts in this place, as well as the junction of the road to Novi Sad to the north and the Hopovo and Irig monastery to the south. It should be mentioned that there is also a hospital in this place, which could be developed into a rehabilitation center. A few kilometers further is the Iriška Tower with a larger capacity accommodation facility that is fully functional. At this location, another sign should mark this route, which would deviate from the general rule of one sign every 5 km.

At kilometer 48, there is a tourist recreation complex on building land, which is the last resort on this route. Then, at the end of the route, is the settlement of Velika Remeta, where such special rest areas are not necessary. The sustainability of all buildings on Fruška Gora and the development of the surrounding places will largely depend on this route. In any case, a feasibility study was prepared in accordance with the current prices on the Serbian market.

The second part of the results is a survey of cyclists. A total of 717 respondents participated in the survey (Table 1), of which 370 were male and were 347 female. These data are particularly encouraging, as some previous studies have shown a significantly higher number of male cyclists than women [40]. The increased number of women on cycles is an indicator of longer stays on long-distance cycles, as it can be assumed that more and more couples are venturing out on such rides. It can also be seen that cyclists are mainly working and well-educated people, which again suggests that long-distance cycling is a choice informed by healthy habits and worldviews [41]. The cyclists came from six countries (Germany, Hungary, the Netherlands, Austria, Romania, and Slovenia), of which the largest proportion came from Germany (217), where the awareness of cycling is very high and, which is one of the leaders in Europe [10].

**Table 1.** Descriptive statistics.

|  | Frequency | Percent |
|---|---|---|
| Male | 370 | 51.6 |
| Female | 347 | 48.4 |
| Total | 717 | 100.0 |
| Under 15 | 1 | 0.1 |
| 16–30 | 148 | 20.6 |
| 31–45 | 282 | 39.3 |
| 46–60 | 257 | 35.8 |
| Over 61 | 29 | 4.0 |
| Total | 717 | 100.0 |
| Ordinary school to eight grade | 1 | 0.1 |
| High school graduate | 102 | 14.2 |
| College | 201 | 28.0 |
| Bachelor's degree | 264 | 36.8 |
| Master's degree | 122 | 17.0 |
| Doctorate | 27 | 3.8 |
| Total | 717 | 100.0 |
| A student | 8 | 1.1 |
| Self-employed | 95 | 13.2 |
| Out of work | 2 | 0.3 |
| Employed | 605 | 84.4 |
| Retired | 6 | 0.8 |
| Something else | 1 | 0.1 |
| Total | 717 | 100.0 |
| Germany | 217 | 30.3 |
| Slovenia | 111 | 15.5 |
| Hungary | 118 | 16.5 |
| The Netherlands | 154 | 21.5 |
| Austria | 100 | 13.9 |
| Romania | 17 | 2.4 |
| Total | 717 | 100.0 |

Source: Authors' research.

The respondents were very excited to ride a completely new section that they knew nothing about and had not heard or read about anywhere. Route F1, which is just one of the eight routes in Serbia's oldest national park, Fruška Gora, connects some of the most interesting natural and man-made sights of the mountain. Hidden in the greenery of a century-old oak forest you will find lakes (Sot, Bruje, Moharač) where you can swim or fish in the summer months. There are caves (Grgurevačka, Perina), medieval monasteries, unique natural sites, the dry spa "Sofijini izvori", the Vrdnik spa, springs, viewpoints, wineries, ethnic houses, various gastronomic offerings, and much more. Table 2 shows some of the most common responses from cyclists who answered the questions on a five-point Likert scale. When asked how they would rate driving on the F1 route overall, the vast majority of respondents, around 96%, replied that they were satisfied. This is a very important statement considering the information in Table 3, where you can see the respondents' answers to the shortcomings they noticed while driving. The biggest drawback was the loss of internet, i.e., the lack of cycle signage, maps, information, etc., so getting around without a guide was very difficult. Therefore, it can be concluded that respondents were very satisfied with all the other benefits of riding in the national park once they rated their experience with the highest marks. It is also interesting to note that the respondents were satisfied with the routes they rode.

**Table 2.** Respondents' experience regarding the F1 route.

| How Would You Rate Your Experience with the F1 Route? | | Frequency | Percent |
|---|---|---|---|
| Valid | Very satisfied | 456 | 63.6 |
| | Satisfied | 239 | 33.3 |
| | Neutral | 22 | 3.1 |
| | Total | 717 | 100.0 |
| **Rate the condition of the bike path you rode on** | | **Frequency** | **Percent** |
| Valid | Very satisfied | 552 | 77.0 |
| | Satisfied | 150 | 20.9 |
| | Neutral | 15 | 2.1 |
| | Total | 717 | 100.0 |
| **Evaluate the condition of the cycling infrastructure (signs, stops for cyclists, service stations, etc.)** | | **Frequency** | **Percent** |
| Valid | Neutral | 54 | 7.5 |
| | Dissatisfied | 477 | 66.5 |
| | Very dissatisfied | 186 | 25.9 |
| | Total | 717 | 100.0 |
| **Evaluate the natural attractions on the route** | | **Frequency** | **Percent** |
| Valid | Very satisfied | 683 | 95.3 |
| | Satisfied | 34 | 4.7 |
| | Total | 717 | 100.0 |
| **Rate the anthropogenic attractions on the route** | | **Frequency** | **Percent** |
| Valid | Very satisfied | 689 | 96.1 |
| | Satisfied | 28 | 3.9 |
| | Total | 717 | 100.0 |
| **Rate the HR on the route** | | **Frequency** | **Percent** |
| Valid | Very satisfied | 659 | 91.9 |
| | Satisfied | 47 | 6.6 |
| | Neutral | 11 | 1.5 |
| | Total | 717 | 100.0 |
| **Rate the state of accommodation facilities on the route** | | **Frequency** | **Percent** |
| Valid | Very satisfied | 120 | 16.7 |
| | Satisfied | 263 | 36.7 |
| | Neutral | 185 | 25.8 |
| | Dissatisfied | 132 | 18.4 |
| | Very dissatisfied | 17 | 2.4 |
| | Total | 717 | 100.0 |
| **Rate the availability of route information** | | **Frequency** | **Percent** |
| Valid | Very dissatisfied | 697 | 97.2 |
| | Dissatisfied | 20 | 2.8 |
| | Total | 717 | 100.0 |

Source: Authors' research.

**Table 3.** Disadvantages of the F1 route.

| What Would You Single Out as the Biggest Drawback of the F1 Route? | Frequency | Percent |
|---|---|---|
| Absence of signs along the route | 90 | 12.6 |
| The paths are not marked | 110 | 15.3 |
| There are no service stations | 125 | 17.4 |
| There are no maps or any information; we would have had a hard time finding our way if it were not for the guide | 147 | 20.5 |
| The internet is lost in some parts, so it is easy to get lost since the routes are not marked | 151 | 21.1 |
| There is no offer of this route on the official EuroVelo website | 90 | 12.6 |
| I cannot rate | 4 | 0.6 |
| Total | 717 | 100.0 |

Source: Authors' research.

In the period from 2021 to 2023, most of the local roads through the national park were repaired and put in order. The road on which the cyclists traveled is a wide, paved municipal road, next to which there is also a hiking and cycling route used by local hikers and cyclists. However, none of these paths are specifically marked and adapted for cyclists. The answer to the next question confirmed the previous assertions, as the cyclists confirmed that the paths, despite being good and pleasant to cycle on, have no content or markings. On the other hand, it is possible to reach natural and anthropogenic sights of exceptional value via the paths, whether cycling on mountain routes through the forest or on asphalted local roads. In addition, the cyclists confirmed that the people they met along the route were extremely friendly, warm, sociable, and hospitable. It was possible to talk to anyone, and everyone they encountered was willing to help. Along the way, they came across a children's resort on Letenka, a mountain lodge on Zmajevac, and the Norcev Hotel on Iriški Venac. All three facilities were rated very positively by the respondents. The negative ratings concerned the number of accommodation facilities, suggesting that there should definitely be more accommodation capacity on each section of the road.

Table 4 shows the main advantage of cycling through the national park and gives reasons why rerouting of the route should be seriously considered. First, most responses refer to the fact that it is nicer to cycle through nature and the forest than through populated areas, especially in the summer (36%). In addition, such an environment encourages people to be extra careful with nature, not to stray from the path, not to litter, and to rest at clearly marked stops and viewpoints (16.5%). They also pointed out that there is not much traffic even on the roads intended for cars and that it is nice and safe to cycle. In addition, the goods in the stores are cheaper than in the cities, and they also like the food, especially in the Banstolka Ethno House, where it is prepared in the traditional way.

Another disadvantage of the F1 route is the fact that this route is not located at the official EuroVelo station, which means that it is impossible to cycle it until it is retraced and officially marked. So, all the negative aspects of the route stem from its absence on the official European Cyclist Transversal cycling map. In fact, the data obtained here show that there should be greener alternatives to the official EuroVelo routes that run on paved cycle paths through villages and towns, at least on the sections where the conditions are right. The benefits of such paths are double-edged. It is more pleasant and beautiful to cycle through nature, which is the advantage of such paths for cyclists; on the other hand, cyclists are a sustainable category of tourists who care about the space they pass through. Movement through the national park would be controlled and plants and animals would be protected from trampling and disturbance.

**Table 4.** Advantages of the F1 route.

| What Would You Single Out as the Biggest Advantage of the F1 Route? | Frequency | Percent |
|---|---|---|
| Beauty of nature | 37 | 5.2 |
| Along the route, numerous wine cellars are open for visits | 29 | 4.0 |
| Less traffic | 28 | 3.9 |
| Beautiful views and viewpoints | 28 | 3.9 |
| The route we drove on was very nicely decorated | 19 | 2.6 |
| People are kinder and more intimate | 22 | 3.1 |
| The food is homemade | 18 | 2.5 |
| It is possible to take a break at each step | 27 | 3.8 |
| There are wonderful monasteries along the route | 29 | 4.0 |
| The national park is taken care of, everything is arranged and clean, and it is really nice to cycle | 118 | 16.5 |
| Along the route, there are lakes where you can refresh yourself | 40 | 5.6 |
| Prices are lower in shops than in cities | 26 | 3.6 |
| A welcome feeling at every stop | 37 | 5.2 |
| It is more pleasant to cycle through the forest in the summer than on concrete | 258 | 36.0 |
| I cannot rate | 1 | 0.1 |
| Total | 717 | 100.0 |

Source: Authors' research.

The results of the survey of the national park rangers revealed a few very important facts about cyclists. It was confirmed to us several times that cyclists do not leave litter behind and also collect the litter that someone else has left behind. This is in line with the views of researchers who say that cyclists are socially responsible and belong to the category of sustainable tourists [42,43]. Cyclists move exactly on designated paths, do not deviate from marked paths, park their two-wheelers properly, and show consideration for the forest and nature. In this way, they represent a desirable category of tourists who are aware of the importance of the symbiosis between man and nature.

The results of the survey we carried out among the entrepreneurs along the F1 cycle route provided us with very interesting results. It should be noted that this is a very small sample, as the F1 route is only part of the routes that can be cycled in the national park. It is a path that runs through the central part of the park, so you have to take other routes to reach populated areas, and there are very few commercial buildings along the way. Everyone we spoke to agreed that traffic increases in the summer months and foreign languages can be heard. The conversation with the president of the Banstolka Women's Association is particularly noteworthy. She told us that the number of visitors triples in the summer months, that many cyclists from all over the world visit her, and that an additional labor force is hired during that time. She also said that her restaurant sells homemade products that are handmade by the people of the surrounding villages. She can then sell whatever the peasants bring her. She particularly points out the women of all the associations who sell their handicrafts as souvenirs and local products, which the cyclists often buy and take home. In this way, the local economy is strengthened and women, a particularly vulnerable category, are empowered [44]. This confirms the hypothesis put forward by Blondiau et al. [28].

## 4. Conclusions

The Fruška Gora National Park has a diverse tourist offer, is close to traditional and emerging tourist markets, has a long history and general recognition, has preserved natural resources, has a good proportion of communication, and has great human potential.

However, the process of transforming comparative into competitive advantages in tourism in the Fruška Gora National Park is part of the reform process, and the political relationship with tourism is an important creator of national prosperity. With regard to the benefits that the national park can derive from the development of cycling, numerous further studies must be carried out in the future, which should cover all possible cycling routes that can be designed (F1–F8, totaling approximately 600 km). By designing, marking, and including them in the official EuroVelo 6 transversal routes, a win–win effect would be achieved, both for the national park itself and for the human factor (cyclists and business people). A particularly sensitive category is rural women, who could be involved in the production of healthy food and handicrafts, thus allowing for cycling tourism to have an empowering effect. This could be the subject of a future study including rural destinations where women's associations similar to Banstolka exist. The women in this ethnic cuisine restaurant were particularly delighted with the fact that a large percentage of the women were cyclists, which further empowered them to offer their products and be part of the development of cycling tourism in the national park. Cycling has an impact on the economy of the destination, as it creates numerous jobs along the route itself and in the region attracted by the route.

Particularly interesting is the conclusion from the survey of Fruška Gora National Park employees, which indicates that cycling is a form of tourism that should be developed in the national park. This applies to tourists who are loyal to the destination, care about the destination, and work and live in accordance with all the principles of sustainability. Based on the "blueprint" of a planned cycle route presented in this paper, it can be concluded that the development of cycling tourism in the destination would bring numerous benefits in terms of investment, construction of the necessary infrastructure and superstructure, signalizations, interactive maps, and marketing.

It was assumed that it would be more pleasant for cyclists to ride through wooded areas, especially in the summer, which turned out to be a correct statement. Despite the fact that the paths are not marked and there is no adequate cycling infrastructure or superstructure, the impression of cycling in the national park was extremely positive. It is therefore necessary to put additional pressure on the state and decision-makers to make the development of cycling tourism a priority. It is a sustainable form of tourism that benefits everyone. It has been shown that cyclists protect nature and that they do not damage the ecosystem or biodiversity. They cycle on the designated paths and stop at the designated places. At the same time, they do not pollute the environment and even collect garbage when they find it. Controlled movement through nature enables a better relationship with nature and a higher level of protection. On the other hand, cyclists spend their money in stores, restaurants, hotels, hostels, ethnic houses, and more. They stop often and spend a lot. They like to interact with the local population, share their experiences, and are very friendly and communicative. They are able to stay an extra day in a place where they feel comfortable and at ease.

One of the main limiting factors is the lack of dedicated research on tracing cycling routes to connect larger areas and promote cycling tourism. However, some researchers provide examples of how to develop cycling tourism in destinations and optimally route cycling routes [37]. In view of this, research is an important contribution to science, especially in the context of future potential investment projects. Following the research of Plzakova [45], it is very important to point out that in tourism, especially in specialized forms of tourism, which includes bicycle tourism, investment through projects is important. Projects are realized when the importance of developing a particular form of tourism is understood. Scholars agree that cycling tourism is a sustainable form of tourism that is desirable for the destination [41]. In the research conducted by Meng and Han [39], it is found that tourists who opt for cycling tourism are those who have a high level of responsibility and awareness of themselves, the destinations, and the importance of sustainable development. On the other hand, Mundet et al. [2] provide a constructive

proposal on how to approach the transformation of space and answer some very important questions about how to develop cycling tourism in destinations, using Spain as an example.

Only one thing can be concluded from all this, namely, that cycling should extend its positive effects to all areas of society. In situations where education is necessary, various workshops can be organized and those who know the most about the positive aspects of the development of cycling can be brought to the workshops. Examples of good practice should be followed and learned from. Only in this way will the future be connected and on two wheels. If tracing the first route proves to be effective, it would enable the entire destination to be networked. Future research could be shifted from the field of economic indicators of sustainability to the sociological field. For example, studies could investigate how cycling affects not only interpersonal communication but also, very importantly, intrapersonal communication, which is reflected in the needs and the possibility of the motive to satisfy them.

**Author Contributions:** Conceptualization, A.V. and M.K.; methodology, A.V.; software, P.B.; validation, A.V., R.B. and M.K.; formal analysis, P.B.; investigation, R.B.; resources, A.V.; data curation, A.V.; writing—original draft preparation, A.V.; writing—review and editing, R.B.; visualization, A.V.; supervision, P.B. All authors have read and agreed to the published version of the manuscript.

**Funding:** This research was funded by The Science Fund of the Republic of Serbia under grant number 7739076.

**Institutional Review Board Statement:** This study was conducted in accordance with the Declaration of Helsinki and approved by the Institutional Ethics Committee of Singidunum University (protocol code: 161; date of approval: 22 February 2024).

**Informed Consent Statement:** Informed consent was obtained from all subjects involved in this study.

**Data Availability Statement:** The aggregated data analyzed in this study are available from the corresponding author upon reasonable request.

**Conflicts of Interest:** The authors declare no conflicts of interest.

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
