# Peer review of "Sustainable Approach to the Development of the Tourism Sector in the Conditions of Global Challenges"

_sustainability, doi:10.3390/su16052098_

Round 1

Reviewer 1 Report (Previous Reviewer 1)

Comments and Suggestions for Authors

Author Response

Respected Reviewer,

Thank you very much for each of your comments. These are constructive comments that help our work to be better and to find its place in the scientific literature. 

Reviewer 2 Report (Previous Reviewer 2)

Comments and Suggestions for Authors

The article has been edited to some extent, but all my comments have not been accepted.

It must be evident from the abstract of the article that it is an abstract of a scientific article. I can repeat: The abstract of the article is insufficient (objective, background, methods, results, conclusion, and application are missing).

The purpose is newly specified in the introduction, but the other recommendations were not accepted.

Not a single change was made in the literature review section.

A fundamental problem is that the paper still does not include a discussion in the context of previously published work.

Future research direction has been added. However, the article still lacks implication and limiting factors.

Author Response

Reviewer 3 Report (Previous Reviewer 3)

Comments and Suggestions for Authors

My comments are as follows.

- What are the limitations of this study?After presenting the theoretical background, the research problem and hypothesis are not clearly stated. Additionally, the conclusion lacks sufficient theoretical and practical implications. Finally, the limitations of the study are not discussed.

Author Response

Reviewer 4 Report (Previous Reviewer 4)

Comments and Suggestions for Authors

This new version is now ready for publication and the summary problem has already been resolved. Congratulations!

Author Response

Respected Reviewer,

Thank you very much for each of your comments. These are constructive comments that help our work to be better and to find its place in the scientific literature. 

Round 2

Reviewer 2 Report (Previous Reviewer 2)

Comments and Suggestions for Authors

Thank you for incorporating my comments, which I believe contributed to improving the quality of the article.

Please do not include abbreviations in the abstract that are not explained.

Reviewer 3 Report (Previous Reviewer 3)

Comments and Suggestions for Authors

Congratulations.

You complemented this paper well.

This manuscript is a resubmission of an earlier submission. The following is a list of the peer review reports and author responses from that submission.

Round 1

Reviewer 1 Report

Comments and Suggestions for Authors

Reviewer 2 Report

Comments and Suggestions for Authors

The article is devoted to the issue of tourism, but I miss the connection with the issue of sustainability. At the same time, the article lacks scientific character.

The abstract of the article is insufficient (objective, background, methods, results, conclusion, and application are missing). The introduction doesn´t contain all required parts (definition of the issue, explanation why the given issue is topical and important, summary of the findings on the given topic to date and the aim of the authors of the article). The literature review section is not well structured and argued. The literature review is not directly related to the research carried out. The research is not related to the issue of sustainability. There is a complete lack of discussion. Implications, limiting factors and future research directions are missing in the conclusion.

Reviewer 3 Report

Comments and Suggestions for Authors

Thank you for giving me the opportunity to review this study. I have reviewed this study very seriously and I leave the following comments.

It is necessary to describe the implications of this study in the Abstract.

The background of the introduction does not support the justification of this study. In other words, the goal of this study, which presents a cycling path change project, and the contents of lines 22 to 49, are judged to be irrelevant. In addition, the review of previous studies related to this study and the differentiation of this study were not presented. Therefore, this study did not provide justification for the purpose.

This study included a theoretical background in the category of the introduction. And the biggest issue did not present the research question for the purpose of this study.

No discussion points were found between previous studies in relation to this study.

This study did not present any theoretical implications, nor did the limitations of this study and future research directions.

Reviewer 4 Report

Comments and Suggestions for Authors

The article presented is really very interesting but lacks scientific rigor in its writing.
The abstract of the article itself needs to be reformulated.
I advise authors to create a new structure for the article. For example, the results chapter begins with a series of results and opinions that we don't know where they come from. Then, they go on to detail all the results of the survey of 717 cyclists. Here the statistical analysis seems more like a list of descriptions without further contributions. Why not study relationships (causal or correlational) between concepts?
In my opinion, this article is currently a re-transcription of a market study and not a scientific article. I invite the authors to rewrite the article for a new review.

Comments on the Quality of English Language

Rewrite the abstract please and restructure your manuscript.
